# Histone Demethylase KDM5C Drives Prostate Cancer Progression by Promoting EMT

**DOI:** 10.3390/cancers14081894

**Published:** 2022-04-08

**Authors:** Anna-Lena Lemster, Elisabeth Sievers, Helen Pasternack, Pamela Lazar-Karsten, Niklas Klümper, Verena Sailer, Anne Offermann, Johannes Brägelmann, Sven Perner, Jutta Kirfel

**Affiliations:** 1Institute of Pathology, University Hospital Schleswig-Holstein, 23538 Luebeck, Germany; anna-lena.lemster@uksh.de (A.-L.L.); helen.pasternack@uksh.de (H.P.); pamela.lazar-karsten@uksh.de (P.L.-K.); verena-wilbeth.sailer@uksh.de (V.S.); anne.offermann@uksh.de (A.O.); sven.perner@uksh.de (S.P.); 2Institute of Pathology, University Hospital Bonn, 53127 Bonn, Germany; elisabeth.sievers@ukbonn.de; 3Department of Urology and Pediatric Urology, University Hospital Bonn, 53127 Bonn, Germany; niklas.kluemper@ukbonn.de; 4Department of Translational Genomics, Faculty of Medicine and University Hospital Cologne, University of Cologne, 50931 Cologne, Germany; johannes.braegelmann@uni-koeln.de; 5Mildred Scheel School of Oncology Cologne, Faculty of Medicine and University Hospital Cologne, University of Cologne, 50931 Cologne, Germany; 6Faculty of Medicine and University Hospital Cologne, Center for Molecular Medicine Cologne, University of Cologne, 50931 Cologne, Germany; 7Institute of Pathology, Research Center Borstel, Leibniz Lung Center, 23845 Borstel, Germany

**Keywords:** prostate cancer, CRPC, histone demethylase, KDM5C, epithelial-to-mesenchymal transition, signaling pathway

## Abstract

**Simple Summary:**

Prostate cancer is the most common cancer in men and is one of the leading causes of cancer-related deaths. During prostate cancer progression and metastasis, the epithelial cells can undergo epithelial–mesenchymal transition (EMT). Here, we show that the histone demethylase KDM5C is highly expressed in metastatic prostate cancer. We establish that stable clones silence KDM5C in prostate cancer cells. Knockdown of KDM5C leads to a reduced migratory and invasion capacity. This is associated with changes by multiple molecular mechanisms. This signaling subsequently modifies the expression of various transcription factors like Snail, Twist, and Zeb1/2, which are also known as master regulators of EMT. Taken together, our results indicate the potential to therapeutically target KDM5C either alone or in combination with Akt/mTOR-inhibitor in prostate cancer patients by targeting the EMT signaling pathways.

**Abstract:**

Prostate cancer (PCa) poses a major public health problem in men. Metastatic PCa is incurable, and ultimately threatens the life of many patients. Mutations in tumor suppressor genes and oncogenes are important for PCa progression, whereas the role of epigenetic factors in prostate carcinogenesis is insufficiently examined. The histone demethylase KDM5C exerts important roles in tumorigenesis. KDM5C has been reported to be highly expressed in various cancer cell types, particularly in primary PCa. Here, we could show that KDM5C is highly upregulated in metastatic PCa. Functionally, in KDM5C knockdown cells migratory and invasion capacity was reduced. Interestingly, modulation of KDM5C expression influences several EMT signaling pathways (e.g., Akt/mTOR), expression of EMT transcription factors, epigenetic modifiers, and miR-205, resulting in increased expression of E-cadherin and reduced expression of N-cadherin. Mouse xenografts of KDM5C knockdown cells showed reduced tumor growth. In addition, the Akt/mTOR pathway is one of the classic signaling pathways to mediate tumor metabolic homeostasis, which is beneficial for tumor growth and metastasis. Taken together, our findings indicate that a combination of a selective KDM5C- and Akt/mTOR-inhibitor might be a new promising therapeutic strategy to reduce metastatic burden in PCa.

## 1. Introduction

Prostate cancer (PCa) is the most commonly diagnosed cancer in men worldwide and remains a leading cause of cancer-related mortality. Most PCa-related deaths are caused by development of metastatic disease and castration resistance. PCa frequently metastasizes to the bones, particularly the spine, where it can lead to pathological fractures with severe consequences. Once the cancer has spread to distant sites, including the bones, it is generally considered incurable. The occurrence of bone metastasis dramatically limits the patients’ quality of life. Although the 5-year survival rate of PCa patients improved in the last ten years and is now at approximately 99% for localized PCa in the United States, this rate decreases to 32% in patients with distant metastasis [1]. The 10-year survival rate is at 18.5% in PCa patients with distant metastasis.

Metastatic spread is a step-wise process that involves loss of intercellular cohesion, cell migration, angiogenesis, access to systemic circulation, survival in circulation, evasion of local immune responses, and growth in distant organs. Cancer cells can accomplish multiple steps of the metastatic process at once through the engagement of a latent cellular program, the epithelial–mesenchymal transition (EMT) [2,3]. The induction of EMT is accompanied by a dynamic reprogramming of the epigenome involving changes in DNA methylation and several post-translational histone modifications. Histone lysine methylation is associated with either gene activation or silencing, depending on the site of the lysine residues. During TGFβ-mediated EMT, there is a global reduction in the heterochromatin mark H3 Lys9 dimethylation (H3K9me2), an increase in the euchromatin mark H3 Lys4 trimethylation (H3K4me3), and an increase in the transcriptional mark H3 Lys36 trimethylation (H3K36me3) [4].

Members of the KDM5 family (also known as JARID1) act as histone H3K4 demethylases. KDM5 family members have various biological functions; they can be crucial in the expression and repression of oncogenes and tumor suppressor genes, and can themselves serve as both. For instance, KDM5B is overexpressed in a variety of cancer types, and high levels of KDM5B have been observed in breast cancer and PCa (reviewed in [5]). Previously, we systematically investigated KDM5C expression patterns in two independent radical prostatectomy cohorts with a total of 761 primary PCas by immunohistochemistry and demonstrated that KDM5C was significantly overexpressed in primary PCa [6]. The oncogenic role of KDM5C was also observed in gastric cancer [7] and hepatocellular carcinoma [8]. Recent evidence indicates that members of the KDM5 cluster may also be involved in PCa metastasis. KDM5B, which is significantly overexpressed in localized and metastatic PCa, is an androgen receptor coactivator and may play an important role in controlling PCa cell invasion and metastasis [5]. It was demonstrated that KDM5D levels were highly downregulated in metastatic PCa. In addition, the KDM5D gene was frequently deleted in metastatic PCa [9]. However, the role of KDM5C in metastatic PCa has not been well studied.

Here, we show that KDM5C expression is enhanced in the metastatic tissue of lymph node-metastasized PCa and castration-resistant PCa (CRPC). Silencing of KDM5C in bone metastatic PCa cell line PC3 inhibited tumor growth in mouse xenografts. In addition, cellular depletion of KDM5C by shRNA inhibited PC3 cell migration, invasion, and epithelial–mesenchymal transition in vitro. Modulation of KDM5C expression influences several EMT signaling pathways (Hedgehog, Wnt, Notch, TGFβ, PI3K-AKT-mTOR), the expression of transcription factors (SNAI2, TWIST1, ZEB1, ZEB2), epigenetic modifiers (DNMT1, EZH2), and miR-205, resulting in increased expression of E-cadherin and reduced expression of N-cadherin.

Our findings provide a novel mechanistic role of KDM5C in PCa metastasis, suggesting that KDM5C may serve as a potential therapeutic target for advanced PCa patients.

## 2. Materials and Methods

### 2.1. Immunohistochemical Analysis and Quantification of Protein Expression

Formalin-fixed and paraffin-embedded prostatic tissue of lymph node metastasized PCa (N = 95) and castration-resistant PCa (CRPC) (N = 28) were provided from cohorts of the University Hospital of Luebeck. Our study was approved by the Ethics Committee of the University of Luebeck (17-313). These tissues and xenograft tumors were used to prepare tissue microarrays (TMA), as previously described in Shaikhibrahim et al. [10]. The following primary antibodies, KDM5C (34718, dilution 1:1000/1:2000, Abcam, Cambridge, UK), E-Cadherin (36, Ventana Medical System, Tucson, AZ, USA), Ki-67 monoclonal rabbit (30-9, Ventana Medical System), were used and detected with Ultra View Universal DAB Detection Kit (Ventana Medical System). Slides were then digitized using the Zeiss Panoramic Midi Scanner (3DHistech, Budapest, Hungary). The images of KI-67 staining were analyzed semi-quantitatively with the software Definiens Tissue Studio 2.1 (Tissue Studio v.2, Definiens AG, Munich, Germany) as previously described in Stein et al. [6]. KDM5C and CDH were analyzed through eyeball analyses, the definition of nuclear KDM5C and membranous CDH1 as negative or positive by two experienced observers. Samples with absence of carcinoma or lack of tissue were excluded.

### 2.2. Cell Culture and Lentiviral Transduction

PC3 cells were cultured in DMEM/F12 medium supplemented with 10% FBS, 100 U/mL Penicillin/Streptomycin, and 1% L-Glutamine (all from Invitrogen, Thermo Fisher Scientific, Waltham, MA, USA).

For lentiviral transduction, five individual clones from MISSION™ shRNA targeting JARID1C NM_001146702 (TRCN 0000358549, TRCN 0000234960, TRCN 0000234961, TRCN 0000022085, TRCN 0000022087) and one shControl clone (SHC004) (all from Sigma-Aldrich, St. Louis, MO, USA) were co-transfected with lentiviral packaging plasmids (psPAX2 plasmid 12,260 and pMD2.G plasmid 12,259 from Addgene, Teddington, UK) into HEK 293T cells. The resulting lentiviral particles were used to infect PC3 cells. Twenty-four h post infection, four µg/mL puromycin was added to select the infected cells. Infection efficiency was approximately 90%. Then, five days post-selection, cells of each clone were isolated by serial dilution in 96-well plates under puromycin selection to obtain single cell clones.

### 2.3. MTT Cell Proliferation Assay

MTT assay was performed, according to the manufacturer’s protocol (Roche, Mannheim, Germany) for the different time periods of 24 h, 48 h, and 72 h.

### 2.4. Migration and Invasion Assays

Cellular motility was analyzed by Transwell migration (Transwell Boyden chamber (#353097, Falcon/Corning, New York, NY, USA)) and Invasion chamber (Matrigel™ Invasion chamber (#354480, BD Biosciences, Heidelberg, Germany)) assays using cells prestarved in 2% FBS medium for 24 h as previously described in Sievers et al. [11].

### 2.5. RNA isolation and Quantitative Real-Time PCR (qRT-PCR)

RNA was extracted using PureLink^®^ RNA Mini Kit (Ambion Life Technologies, Thermo Fisher Scientific), according to the manufacturer’s protocol. Afterwards, cDNA synthesis was performed with 1 μg RNA using SuperScript III and Oligo (dT) 12-18 primers (Invitrogen, Thermo Fisher Scientific). Gene expression was quantified by real-time PCR as described by Lim et al. [12]. Each sample was run in triplicate and relative expression was determined by normalization to the TATA-binding protein (TBP) or Hypoxanthine-Guanine Phosphoribosyl Transferase (HPRT) using the 2^−ΔΔCt^ method. Error bars indicate standard error of the mean (SEM). Primer sequences are available in the Appendix A.

### 2.6. Isolation and TaqMan PCR Analyses of miRNA

MicroRNA from sh-transfected KDM5C knockdown and control cells was extracted using Invitrogen’s mirVana miRNA Isolation Kit (#AM1560, Thermo Fisher Scientific), according to the manufacturer’s specifications. Specific TaqMan™ advanced miRNA assays (TaqMan™ advanced miRNA cDNA Synthesis Kit (#A25576) and TaqMan™ Fast advanced Master Mix Kit (#4444556)) for the detection of hsa-miR-205-5p (#477967 mir) and the detection of endogenous control hsa-miR-361 (#478056 mir) were purchased from Invitrogen (Thermo Fisher Scientific) and used according to the manufacturer’s specification. Each sample was run in triplicate in a 10 μL reaction. Relative expression was determined by normalization to endogenous miRNA control using the 2^−ΔΔCt^ method. Error bars indicate SEM.

### 2.7. Protein Extraction and Western Blot Analyses

Protein lysates were extracted from cells and blotted as described in Schulte et al. [13]. The following antibodies were used: KDM5C/JARID1C (1:1000, ab34718, Abcam); β-ACTIN (1:10.000, A5441; Sigma-Aldrich); E-cadherin (1:1000, #3195, Cell Signaling, Cambridge, UK); N-cadherin (1:1000, #14215, Cell Signaling); SNAI2/Slug (1:1000, #9585, Cell Signaling); α-TUBULIN (1:1000, #2144, Cell Signaling); SMAD1 (1:1000, #6944, Cell Signaling); SMAD4 (1:1000, #38454, Cell Signaling); H2Aub, (1:1000, #8240, Cell Signaling); H3K4me2, (1:1000, #035050, Diagenode); H3K4me3 (1:1000, ab1012, Abcam); STAT6 D3H4 (1:1000, #5397, Cell Signaling); Phospho-Smad1 Ser463/465/Smad5 Ser463/465/Smad9 Ser463/465 D5B10 (1:1000, #13820, Cell Signaling).

### 2.8. Kinase Activity Profiling

Tyrosine (PTK) and serine-threonine (STK) kinome activity profiling was performed using a PamStation^®^12 (PamGene, BJ′s-Hertogenbosch, The Netherlands), according to the standard protocol provided by PamGene. Signal intensities for each peptide were analyzed with the PamGene BioNavigator Analysis software tool.

### 2.9. Growth of Xenograft Tumors in Mice

Mouse experiments were performed by Charles River Discovery Research Services Germany GmbH, Freiburg, Germany. PC3 wild-type cells were grown in RPMI 1640 with 10% FCS and 0.05 mg/mL gentamycin. Stable KDM5C knockdown cells were grown in DMEM high glucose with 10% FCS, 0.05 mg/mL gentamycin, and 200 mM L-glutamine. For each cell clone, 10 male NOD SCID mice were injected unilaterally into the flank with 5 × 10^6^ cells in 100 µL volume (50% Matrigel). After injection, an aliquot of the cell suspension was used to verify the cell viability. Mice were weighed for 29 days, and tumors were measured twice a week. Mice were sacrificed and all tumors were collected, fixed in formalin, and embedded into paraffin.

### 2.10. Statistical Methods

For statistical analyses, unpaired Student’s *t*-tests were performed. Results were considered significant when *p*-values were *p* < 0.001 ***; *p* < 0.01 **; *p* < 0.05 *, and n.s. for not significant. Functional protein association networks were generated in STRING DB V11.0 (https://string-db.org/, accessed on 8 September 2021). Network plots were generated in Cytoscape 3.8.1 (http://cytoscape.org/, accessed on 8 September 2021) and analyzed using Network Analyzer V4.4.6 and Omics Visualizer V1.3.0 (Cytoscape App Store-Omics Visualizer).

## 3. Results

### 3.1. KDM5C Expression Is Enhanced in Metastatic Prostate Cancer

Previously, we demonstrated that KDM5C was significantly overexpressed in primary PCa [6]. Here, we analyzed KDM5C expression in a PCa progression cohort. Both lymph node metastases and distant bone metastases showed nuclear KDM5C expression (Figure 1); however, distant metastases showed a much higher nuclear positivity. Of 95 cases of lymph node metastasis, 34% showed a nuclear KDM5C staining. Of 28 cases of distant metastasis, up to 75% showed a strong nuclear KDM5C staining.

### 3.2. KDM5C Knockdown Reduces Cell Migration and Invasion

To further study KDM5C expression different PCa cell types were used. KDM5C is strongly expressed in CRPC cells (PC3) and moderately expressed in metastatic androgen-sensitive cells (LNCaP), whereas KDM5C expression is low in benign prostatic BPH-1 cells (Figure 2A, uncropped Western blot in Appendix A).

Due to the lack of a selective and specific KDM5C inhibitor, we investigated the functional significance of KDM5C in PCa cells by downregulating KDM5C expression in PC3 cells (derived from a hormone-refractory PCa bone metastasis) using lentiviral short hairpin RNA (shRNA) technology (Figure 2A). Four single-cell derived clones showed particularly efficient knockdown of KDM5C. Two of them (PC3 shKDM5C 1.6 and 4.2), as well as two GFP control clones (shControl 1 and 4), were studied in more detail. The KDM5C gene expression was significantly reduced in the KDM5C knockdown cells compared to PC3 wild-type cells. KDM5C expression in shControl clones was not altered (Figure 2B).

As an initial step toward understanding the role of KDM5C in PCa events, we determined whether KDM5C expression alters cell phenotype and regulates PCa cell proliferation, migration, and invasion. Knockdown of KDM5C results in a minor impairment of proliferation measured by viability assay (Figure 2C). The phenotype of the KDM5C knockdown cells varied from that of the control cells (Figure 2D), and the KDM5C knockdown cells adhered less to the flask compared with control cells. Global levels of histone modifications were not affected by the knockdown of KDM5C (Appendix A, uncropped Western blot in Appendix A). This indicates its function through gene locus specific modifications.

Next, we investigated the migration and invasion capacity of KDM5C knockdown and control cells. Knockdown of KDM5C significantly reduces migration and invasion ability by approximately 75% (Figure 2E,F). These results suggest that KDM5C may act as an epigenetic modulator of migration and invasion.

### 3.3. KDM5C-Mediated Changes in Epithelial-Mesenchymal Transition (EMT)

To investigate mechanisms by which KDM5C expression affects cell migration and invasion, we explored the effect of KDM5C knockdown on the expression of genes involved in EMT. The mesenchymal cell–cell adhesion molecule N-cadherin and SNAI2 promote EMT. In contrast, the epithelial cell–cell adhesion molecule E-cadherin inhibits this process. Knockdown of KDM5C led to increased expression of E-cadherin shown by qPCR (*CDH1*, Figure 3A) and Western blot (Figure 3D, uncropped Western blot in Appendix A). Knockdown also led to a reduced expression of N-cadherin (qPCR: *CDH2*, Figure 3B; Western blot: Figure 3D, uncropped Western blot in Appendix A). The qPCR (Figure 3C) and Western blot (Figure 3D, uncropped Western blot in Appendix A) showed reduced expression of SNAI2 after knockdown of KDM5C. Lymph nodes of PCa patients showed a negative correlation between KDM5C and E-cadherin expression (Appendix A). KDM5C expression was increased in the invasive tumor front whereas E-cadherin expression was reduced in this region.

Several transcription factors respond to microenvironmental stimuli and function as molecular switches for the EMT program. We examined the transcription factors *TWIST1*, *TCF4*, *ZEB1*, and *ZEB2*, as well as *DNMT1* and *EZH2* by qPCR, and found that expression of all factors decreased significantly after knockdown of KDM5C (Figure 4A–F). Transcription factors can regulate EMT on the gene level. In addition, EMT is also regulated on the level of RNA. In particular, non-coding RNAs, known as microRNAs (miRNAs), can either exert a positive or a negative regulatory effect on EMT. In PCa, miR-205 exerts a tumor-suppressive effect by counteracting the epithelial-to-mesenchymal transition and reducing cell migration/invasion [14]. PC3 cells, known for their high metastatic ability, display low expression of miR-205. Knockdown of KDM5C leads to a robust increase in miR-205 (Figure 4G).

### 3.4. KDM5C-Mediated Changes in Signaling Pathways

To determine whether KDM5C knockdown leads to changes in signaling pathways, we first analyzed the Wnt signaling pathway. While the ligands *WNT3A*, *WNT5A*, *WNT7A*, and *WNT11* of the Wnt pathway were significantly upregulated in KDM5C knockdown clones (Figure 5A–D) the receptor *FZD9* (Figure 5E) and the mediator *AXIN2* (Figure 5F) were significantly downregulated.

The Notch signaling pathway is composed of four receptors (NOTCH1, 2, 3, 4) and five ligands (JAG1 and 2 and Delta-like 1, 3, and 4) [15]. Ligand *JAG1* (Figure 5G) and receptor *NOTCH1* (Figure 5H) were significantly downregulated after knockdown of KDM5C. The target gene STAT6 was also significantly downregulated after KDM5C knockdown, as shown by qPCR (Figure 5I) and Western blot (Figure 5J, uncropped Western blot in Appendix A).

Hedgehog pathway ligands, *SHH* and *DHH*, were significantly downregulated in KDM5C knockdown clones (Appendix A). Mediators *GLI1*, *GLI2*, *GLI3*, and *HHIP* were also significantly downregulated in KDM5C knockdown clones (Appendix A).

Finally, we examined mediators and ligands involved in the TGFβ signaling pathway. The ligands *BMP6* and *BMP7* were significantly upregulated (Appendix A) whereas the ligand TGFB1 was significantly downregulated in stable KDM5C knockdown clones (Appendix A). Knockdown of KDM5C resulted in significant upregulation of the mediator *SMAD1* (Appendix A) and significant downregulation of the mediators *SMAD2*, *SMAD4*, and *SMAD7* (Appendix A). Protein expression of SMAD1 was upregulated and SMAD4 was downregulated in knockdown clones (Appendix A, uncropped Western blot in Appendix A).

### 3.5. KDM5C Knockdown Changes Kinase Activity Profile

We used PamGene’s functional kinase assay to measure peptide phosphorylation by protein kinases in KDM5C knockdown and control cells. To measure the tyrosine (Tyr) and serine/threonine (Ser/Thr) kinase activity, we used the PamChips PTK (for Tyr) and STK (for Ser/Thr). Equal amounts of cell lysate from knockdown clones (shKDM5C 1.6 and shKDM5C 4.2) and control clones (shControl 1 and shControl 4) were analyzed. The mean signal intensity of each bait peptide of the stable knockdown clones was normalized to the corresponding peptide of the control clones and expressed as log2 fold change (LFC). We performed hierarchical clustering using Euclidean distance metrics for a graphical heat map (Figure 6A). We also determined the grouped *p*-values of the two stable knockdown clones shKDM5C 1.6 and shKDM5C 4.2 versus the grouped control clones by 2-grouped comparison (unpaired *t*-test) in the BioNavigator software. Figure 6B shows these results in volcano plots (PTK on the left side and STK on the right side) and Figure 6C shows a pathway analysis of the results in a high confidence interaction network.

Largely, the intensity of tyrosine substrate peptide phosphorylation was markedly reduced for most peptides in the knockdown clones. The intensity of phosphorylation was also decreased for most Ser/Thr peptide substrates (Figure 6A–C). We found that the activity of the kinases RPS6, PDPK1, PRKCB, MTOR, and PIK3R1 was reduced in KDM5C knockdown cells (Figure 6C). These are kinases that belong to the mechanistic target of the rapamycin (mTOR) signaling pathway. We also found kinases of the phosphoinositide-3-kinase (PI3K) signaling pathway that had an altered activity in KDM5C knockdown cells (Figure 6C), namely CDK2, CDK4, CDKN1A, RPS6, FOXO3, BAD, EPHA2, PDPK1, CREB1, ERBB2, MTOR, NOS3, PDGFRB, PIK3R1, NFKB1, JAK1, KIT, KDR, EPOR, CSF1R, VTN, and GYS2.

We also used a phosphokinase array with 37 different kinase phosphorylation sites as an independent experiment to investigate the kinase activity. There, we confirmed that CREB1 activity was not only downregulated in the PamGene’s functional kinase assay but also in this independent experiment. We additionally identified a difference in three kinase phosphorylation sites in KDM5C knockdown clones compared to the PC3 wild type and the sh-transfected control 4. Knockdown of KDM5C resulted in decreased phosphorylation of GSK-3α/β (S21/S9), PLC-γ1 (Y783), and RSK1/2 (S221/S227) (Appendix A).

### 3.6. KDM5C Knockdown Reduces Tumor Growth in Mice

To investigate KDM5C knockdown in mouse model, xenotransplants of PC3 and sh-transfected cells were studied. The growth of tumors with PC3 xenografts was significantly faster than the tumor growth of KDM5C knockdown cell xenografts over the course of 29 days (Figure 7A). Figure 7C shows exemplary samples of the xenograft tumors. Hematoxylin and eosin staining showed no morphologic differences in tissue. E-cadherin expression was high and N-cadherin expression was low in KDM5C knockdown xenograft, whereas the Ki-67 proliferation index was not altered (Figure 7B). Expression of KDM5C, CDH1 and CDH2 in xenografts was analyzed by qRT-PCR (Appendix A).

## 4. Discussion

Epithelial-to-mesenchymal transition (EMT) and its reverse mesenchymal-to-epithelial transition (MET) are paramount to the metastatic spread of carcinomas. There is a gradual transition between these phenotypic states. Therefore, carcinoma cells often exhibit a spectrum of epithelial/mesenchymal phenotype(s) [16,17,18].

KDM5C is significantly upregulated in primary [6] and metastatic PCa. Silencing of KDM5C in PCa cells inhibited cell migration and invasion. Knockdown of KDM5C results in modulation of several EMT-associated signaling pathways, thereby decreasing EMT-promoting factors and enhancing MET-promoting factors. Our findings are summarized in a model presented in Figure 8.

An integrated and complex signaling network including transforming growth factor β (TGFβ), Wnt, Notch, and Hedgehog leads to the induction of EMT [19,20,21,22].

TGFβ is known to induce EMT in PCa [23], and BMP signaling, in turn, is involved in both EMT and MET [24]. We demonstrated that the expression of TFGB1, SMAD2, and SMAD4 (proteins of the TGFβ signaling [25]) is reduced and the expression of BMP6, BMP7, and SMAD1 (proteins of the BMP signaling pathway [25]) is increased after KDM5C knockdown. In epithelial kidney cells, BMP6 and BMP7 counteract TGFβ-induced EMT by reinducing E-cadherin [26,27]. In PCa, Buijs et al. showed that BMP7 expression is inversely related to tumorigenic and metastatic potential [28]. SMAD7 is part of a negative feedback loop regulating TGFβ signaling. Increased TGFβ signaling correlates with increased expression of SMAD7 [29]. Knockdown of KDM5C leads to decreased TGFβ signaling and decreased expression of SMAD7.

Wnt proteins can utilize signal transduction through the canonical β-catenin-dependent pathway and the noncanonical β-catenin-independent pathway, both of which are closely associated with EMT [30]. In PCa, WNT5A induced bone metastasis, and increased the activation of the WNT5A/FZD2 pathway leading to enhanced EMT [31,32]. We demonstrated that canonical Wnt ligands WNT3A and WNT7A and non-canonical Wnt ligands WNT5A and WNT11 are upregulated after KDM5C knockdown. However, the Wnt receptor FZD9, which is part of the canonical and non-canonical pathways, and AXIN2 are downregulated by KDM5C knockdown. AXIN2 serves as a negative regulator of the canonical Wnt signaling pathway and may act as a negative feedback loop to suppress the signaling pathway [33]. In addition, inhibition of FZD7 leads to decreased AXIN2 expression [34].

The Notch signaling pathway also participates in EMT [19], and its inhibition partially reverts EMT in lung adenocarcinoma cells and reduces their invasive behavior [35]. Silencing of the receptor NOTCH1 inhibits PCa cell invasion [36], whereas expression of the ligand JAG1 is associated with PCa progression, metastases, and recurrence [37]. We demonstrated that JAG1, NOTCH1, and STAT6 are downregulated by KDM5C silencing.

Several studies show that Shh signaling is associated with EMT and that inhibition of the signaling pathway results in suppression of EMT in PCa [38,39,40]. In the absence of Hedgehog pathway ligands, SHH, DHH, and Indian hedgehog (IHH), the pathway is in an “off” state [22]. We showed that SHH and DHH are downregulated after knockdown of KDM5C. In the presence of ligands, the pathway would be switched “on”, resulting in the activation of the GLI family of transcription factors (GLI1, GLI2, and GLI3). GLI1 functions as a transcriptional activator, while GLI2 and GLI3 can be processed into transcriptional activators or repressors [41,42]. Silencing of KDM5C reduced the steady-state level of GLI1, GLI2, and GLI3. HHIP is a negative feedback regulator of hedgehog signaling [43] and was downregulated in our studies after KDM5C knockdown.

These findings strengthen the idea of a role for KDM5C in the regulation of TGFβ, Wnt, Notch, and Hedgehog leading to EMT.

We investigated the kinase activity profile of KDM5C knockdown cells compared to control cells. The PI3K and mTOR signaling pathways play an important role in the induction of EMT [44,45] and influence EMT in PCa [46,47]. We showed that the activity of several kinases e.g., CDK2, CREB1, RPS6, and mTOR is reduced in KDM5C knockdown cells. Other studies show that CREB1 is upregulated in metastases and its inhibition could reduce cell proliferation [48]. Inhibition of CDK2 is able to reduce cell invasion [49] and shows inhibitory effects on EMT [50], and suppression of RPS6 increases radiosensitivity in PCa [51]. Our phosphokinase array also showed that the activity of CREB1 and additionally GSK-3α/β, PLC-γ1, and RSK1/2 was downregulated in stable KDM5C knockdown clones. These kinases all play a role in tumors’ metastasis [52,53,54,55] and may also play a role in the PI3K–Akt signaling pathway [56,57,58,59,60]. Therefore, a combination therapy with inhibition of KDM5C and the PI3K-Akt-mTOR signaling pathway might be beneficial for the patient.

All of the above-mentioned signaling pathways affect the expression of transcription factors and miRNAs, which then induce or repress E-cadherin and N-cadherin expression. The main function of EMT-related transcription factors is to repress epithelial-associated genes and induce mesenchymal genes [61]. Expression of the master EMT transcription factors is activated early in EMT [17,18]. SNAI1 and ZEB1 are expressed in epithelial cells and SNAI2 and ZEB2 are enriched in the early hybrid EMT state [18]. We could show that expression of ZEB1, SNAI2, and ZEB2 is downregulated in KDM5C knockdown cells. Decreased expression of these transcription factors is likely a consequence of the decreased activity of the above-mentioned signaling pathways in KDM5C knockdown. SNAI2 regulates cell proliferation and invasiveness in PCa through EMT [62]. ZEB1 and ZEB2 drive EMT by downregulation of E-cadherin in PCa [63]. TWIST1, which is also downregulated in KDM5C knockdown cells in our studies, suppresses E-cadherin and induces N-cadherin expression independently of SNAIL proteins [64]. In PCa, TWIST1/androgen receptor signaling mediates crosstalk between castration resistance and EMT [65]. TCF, DNMT1, and EZH2 are indirect repressors of E-cadherin, mediated by expression of ZEB1, ZEB2, and SNAI2, respectively [66,67,68]. We demonstrated that the expression of TCF4, DNMT1, and EZH2 is downregulated upon knockdown of KDM5C.

miR-205 also regulates EMT by targeting ZEB1 and ZEB2 [69] and inhibits cancer cell migration and invasion in PCa [70]. It has been observed that miR-205 is especially down-regulated in cells that have undergone EMT. This is accompanied by a pronounced decrease in E-cadherin and an increase in N-cadherin [69]. Inversely, ectopic expression of miR-205 in mesenchymal cell-initiated MET goes along with upregulation of E-cadherin and reduction of cell locomotion and invasion [69]. Knockdown of KDM5C results in an upregulation of miR-205 in our studies.

Pastushenko et al. discussed that co-expression of epithelial (e.g., E-cadherin) and mesenchymal markers (e.g., SNAI1) rather than either epithelial or mesenchymal states have been associated with poor clinical prognosis [71]. Simeonov et al. showed that in patients with epithelial EMT state, early EMT state and mesenchymal EMT state gene clusters had no association with disease prognosis, however, patients who had enriched late EMT state had a significantly increased risk of death [18]. In pancreatic cancer, the TGFβ signaling pathway is enriched in the late hybrid EMT state and tapered off in the mesenchymal state. Wnt and Notch are enriched only in the late hybrid EMT state and Hedgehog is enriched only in the mesenchymal state. The mTOR signaling pathway is enriched in the transition from the early to the late hybrid EMT state [18]. We found that these pathways were all active in PC3 cells and activity was downregulated in KDM5C knockdown cells. We demonstrated that reduction in viability of KDM5C knockdown cells is only minor in vitro. Therefore, the reduced tumor growth in mouse xenografts is not a result of the differential proliferation of these cells as shown by the unaltered Ki-67 expression in PC3 and KDM5C knockdown cells. Tumor growth might be reduced due to changes in the tumor microenvironment after KDM5C knockdown. Activation of the NF-kappaB pathway in the tumor milieu favors tumor survival and drives abortive activation of immune cells [72]. This pathway might be less active in KDM5C knockdown cells because NF-kappaB (NFKB1) activity was downregulated in these cells.

## 5. Conclusions

We found that the signaling pathways of the late EMT states are activated in PC3 wild-type and reduced after knockdown of KDM5C. This results in increased expression of E-cadherin, and a decreased expression of N-cadherin. The cells migrate less and are less invasive than PC3 wild-type cells. KDM5C knockdown also leads to reduced tumor growth in in vivo mouse xenografts. Therefore, we hypothesize that inhibition of KDM5C by a selective compound will hinder the process of EMT and delay progression of PCa.

## Figures and Tables

**Figure 1 cancers-14-01894-f001:**
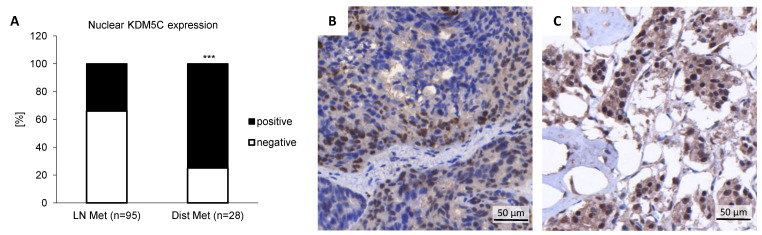
Immunohistochemical staining of KDM5C in a prostate cancer progression cohort. (**A**) Nuclear KDM5C expression in lymph node (LN) metastasis and bone (Dist) metastasis; (**B**) Representative sample of KDM5C protein expression in an invasive tumor edge in lymph node metastasis; (**C**) Representative sample of KDM5C protein expression in bone metastasis showing strong nuclear staining in tumor cells. (*** = *p* < 0.001).

**Figure 2 cancers-14-01894-f002:**
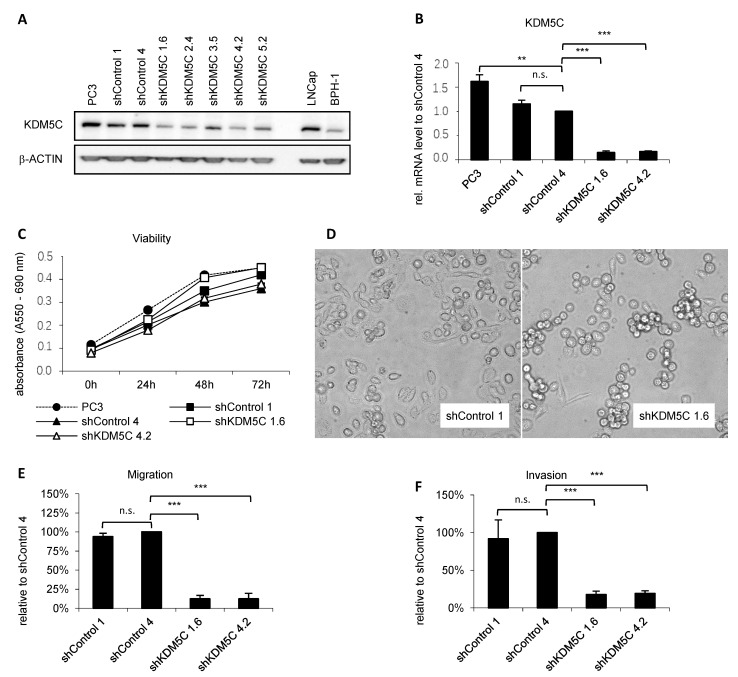
KDM5C expression after shRNA-mediated knockdown. (**A**) Representative Western blot of KDM5C expression in prostate cancer cell line PC3, different shRNA-transduced cell clones, LNCap, and BPH-1. ShControl clones with GFP serve as a control for the lentiviral transduction procedure. The loading control is β-ACTIN; (**B**) qRT-PCR analysis of KDM5C mRNA expression in prostate cancer cell line PC3 and different shRNA-transduced GFP control as well as KDM5C knockdown clones. mRNA expression was normalized to the nuclear housekeeper TBP and in relation to the shControl 4 clone. Error bars represent standard error of the mean from three independent experiments; (**C**) Viability of PC3 cells and several shRNA-transduced cell clones. One representative MTT assay of the parental cell line PC3 and its derived shRNA-controlled clones. Error bars represent standard error of the mean from the quintuplicates; (**D**) Phenotype of shControl clones compared to KDM5C knockdown clones. KDM5C knockdown cells were less adherent to the flask compared with control cells (15× magnification); (**E**) Migration and (**F**) Invasion of PC3 derived shRNA-transduced cell clones tested by Boyden and Invasion chamber assays. Error bars represent standard error of the mean from three independent experiments. (n.s. = not significant; ** = *p* < 0.01; *** = *p* < 0.001).

**Figure 3 cancers-14-01894-f003:**
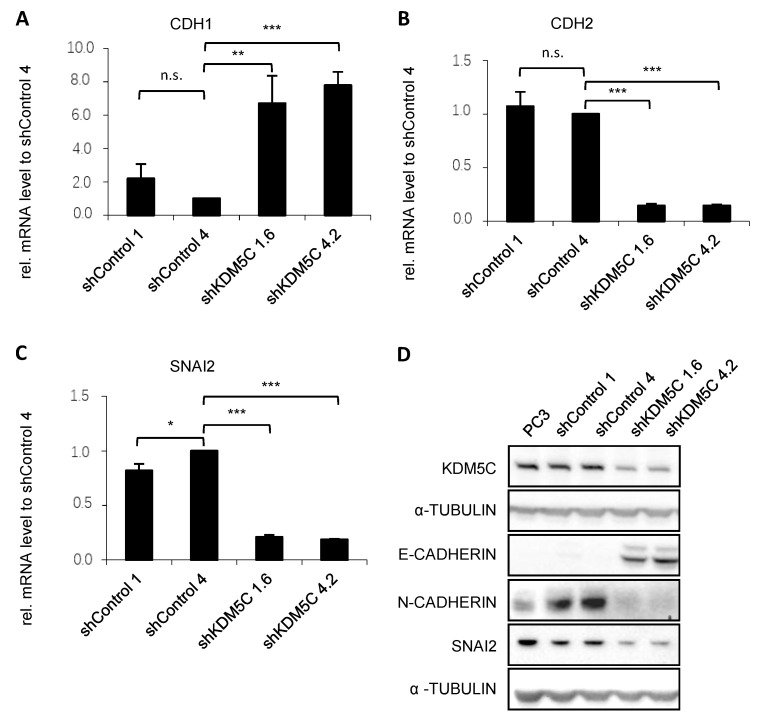
Differential expression of epithelial and mesenchymal markers ((**A**) CDH1; (**B**) CDH2; and (**C**) SNAI2) in PC3 and shRNA-transduced cell clones. qRT-PCR analysis of mRNA expression in GFP control as well as KDM5C knockdown clones. mRNA expression was normalized to the nuclear housekeeper *TBP* in (**A**,**C**) and *HPRT* in (**B**) and in relation to the shControl 4. Error bars represent standard error of the mean from three independent experiments. (**D**) Representative Western blot of the expression of KDM5C and EMT markers in PC3 cells and the shRNA-transduced cell clones. Western blot shows the expression of epithelial marker E-CADHERIN (CDH1) and its transcriptional repressor SNAI2 as well as the typical mesenchymal marker N-CADHERIN (CDH2) in GFP control and KDM5C knockdown clones. α-TUBULIN serves as a loading control. (n.s. = not significant; * = *p* < 0.05; ** = *p* < 0.01; *** = *p* < 0.001).

**Figure 4 cancers-14-01894-f004:**
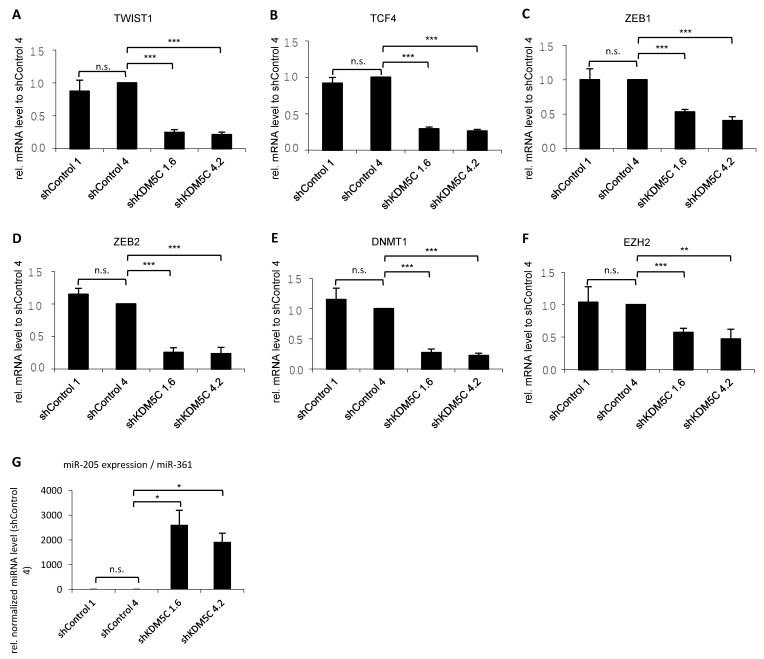
Expression of transcription factors ((**A**) *TWIST1*; (**B**) *TCF4;* (**C**) *ZEB1*; (**D**) *ZEB2*); epigenetic regulators (**E**) *DNMT1;* and (**F**) *EZH2*); and miR-205 (**G**), in stable KDM5C knockdown clones. qRT-PCR analysis of mRNA expression (**A**–**G**) in GFP control as well as KDM5C knockdown clones. mRNA expression (**A**–**F**) and microRNA expression (**G**) were normalized to the nuclear housekeeper *TBP* and to miR-361, respectively, and in relation to the shControl 4 clone. Error bars represent standard error of the mean from three independent experiments. (n.s. = not significant; * = *p* < 0.05; ** = *p* < 0.01; *** = *p* < 0.001).

**Figure 5 cancers-14-01894-f005:**
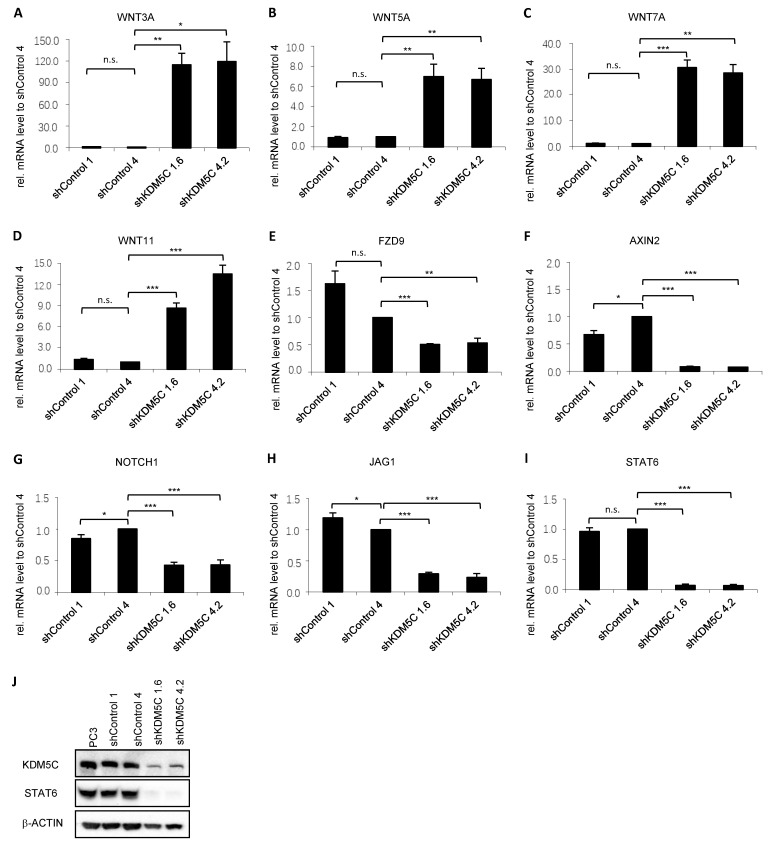
Expression of receptors, ligands, mediator, and target gene of the Wnt (**A**–**F**) and the Notch (**G**–**J**) signaling pathways associated with EMT progression. qRT-PCR analysis of mRNA expression in GFP control as well as KDM5C knockdown clones. Important ligands ((**A**) *WNT3A;* (**B**) *WNT5A;* (**C**) *WNT7A;* and (**D**) *WNT11*); receptor ((**E**) *FZD9*); and mediator ((**F**) *AXIN2*) for the Wnt signaling were examined. For the Notch signaling important receptor ((**G**) *NOTCH1*), ligand ((**H**) *JAG1*), and its target gene ((**I**) *STAT6*) were examined. mRNA expression was normalized to the nuclear housekeeper *TBP* and in relation to the shControl 4 control clone. Error bars represent standard error of the mean from three independent experiments. (**J**) Representative Western blot of the expression of STAT6 in PC3 cells and the shRNA-transduced cell clones. β-ACTIN serves as a loading control. KDM5C shows the knockdown of KDM5C in shKDM5C 1.6 and shKDM5C4.2. (n.s. = not significant; * = *p* < 0.05; ** = *p* < 0.01; *** = *p* < 0.001).

**Figure 6 cancers-14-01894-f006:**
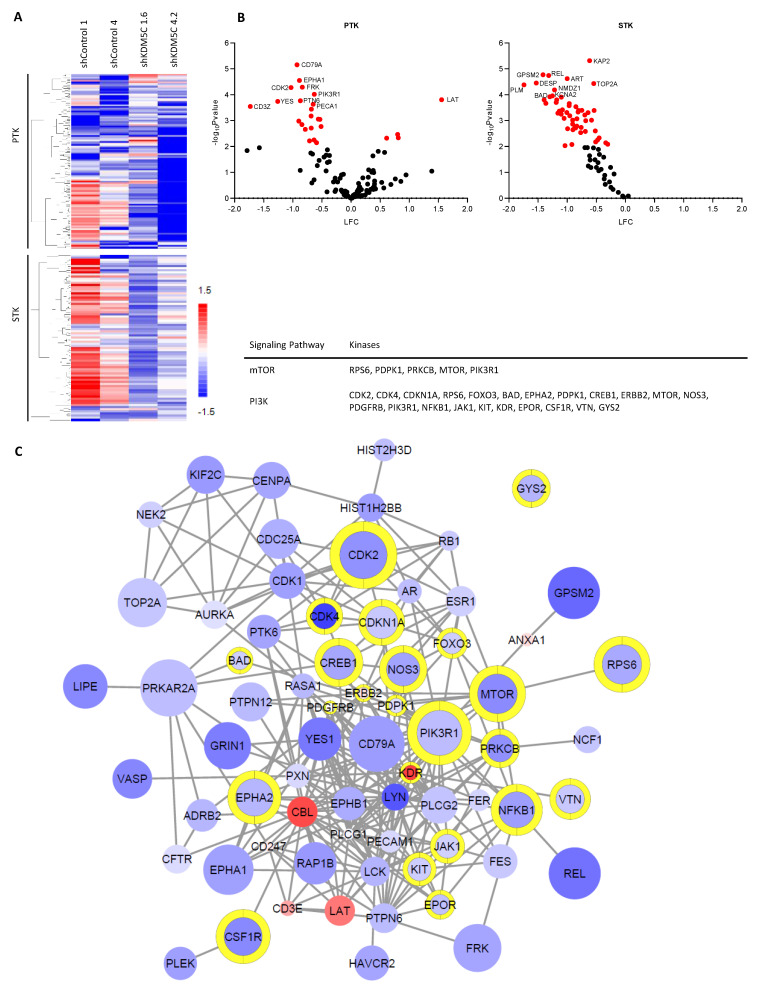
Kinase activity in KDM5C knockdown cells compared to control cells. (**A**) Clustered heat map of increased and decreased phosphorylation of Tyr (PTK, top) and Ser/Thr (STK, bottom) bait peptides in control cells and stabile knockdown of KDM5C; (**B**) Volcano plots of the bait peptide phosphorylation from PTK (**left**) and STK (**right**) were assessed by student’s *t*-test; (**C**) Pathway analysis of peptides from (**B**) (*p* > 0.05) in a high-confidence interaction network (edges) from STRING DB. Node color indicates increased and decreased phosphorylation of Tyr (PTK) and Ser/Thr (STK) bait peptides. Node size indicates −log_10_(*p* value). Yellow border color indicates that the protein is involved in the PI3K-Akt-mTOR signaling pathway. KCNA1, KCNA3, RYR1, PPP1R1A, CACNA1C, GRIK2, DSP, FXYD1, TH, KCNA2, GPR6, ENO2, EPB42, STMN2, PFKFB3, SCN7A, GABRB2, EFS, PHKA1, and MYBPC3 are not shown in the network because they showed no interaction with the kinases in the network.

**Figure 7 cancers-14-01894-f007:**
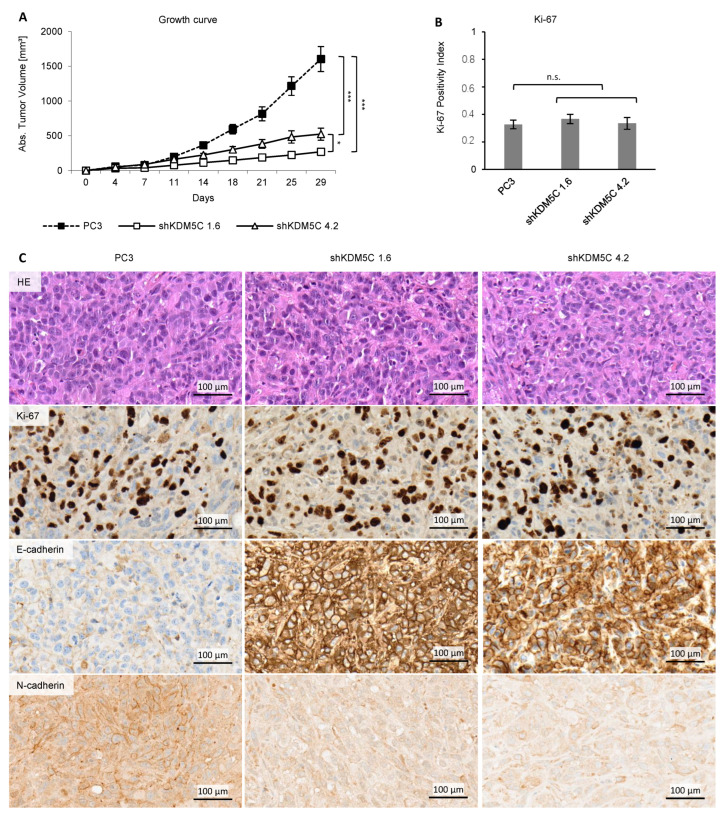
Comparison of mouse xenografts of PC3 wild type and KDM5C knockdown cells. (**A**) Growth curve of PC3 and shRNA-transduced cell tumors in mice xenografts. Error bars represent standard error of the mean from ten independent experiments. Tumors in mice receiving KDM5C knockdown cells were smaller than tumors of wild-type cells; (**B**) Index of Ki-67 positive cells in mice xenografts with PC3 and shRNA-transduced cells. Error bars represent standard error of the mean from eight independent experiments; (**C**) Representative samples of hematoxylin-eosin (HE) staining, as well as Ki-67, E-cadherin and N-cadherin protein expression in PC3 and shRNA-transduced cell mice xenografts. (n.s. = not significant; * = *p* < 0.05; *** = *p* < 0.001).

**Figure 8 cancers-14-01894-f008:**
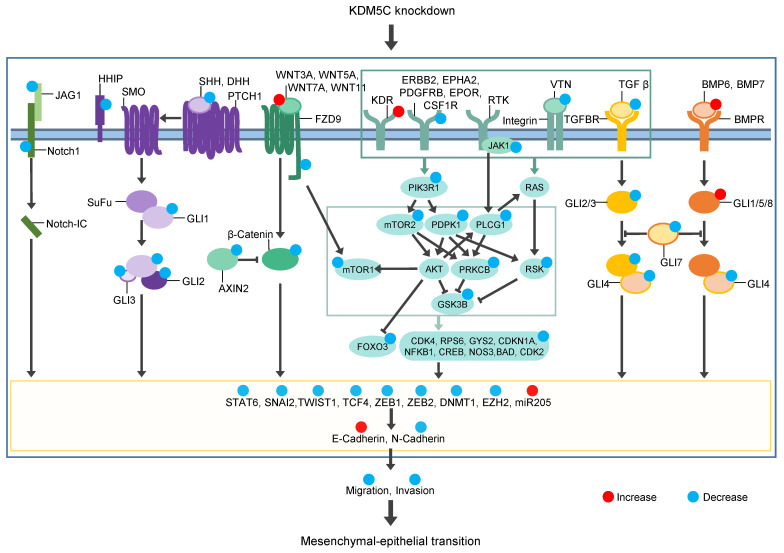
Simplified model of our results on the role of KDM5C in prostate cancer. KDM5C knockdown leads to downregulation of signaling pathways and transcription factors involved in EMT. E-cadherin expression is increased, and N-cadherin expression is decreased after KDM5C knockdown which results in MET.

## Data Availability

All datasets generated and analyzed during the current study are available from the corresponding author on request.

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
