# Peer review of "Histone Demethylase KDM5C Drives Prostate Cancer Progression by Promoting EMT"

_cancers, 2022, doi:10.3390/cancers14081894_

Round 1

Reviewer 1 Report

The authors show that the previously reported prognostic marker of PCa, histone demethylase KDM5C is highly expressed in metastatic prostate cancer. They establish stable clones silencing KDM5C in prostate cancer cell line, PC3. Knockdown of KDM5C leads to a reduced migratory and invasion capacity. This is associated with changes by multiple molecular mechanisms. This signaling subsequently modifies the expression of various transcription factors like Snail, Twist, Zeb1/2, which are also known as master regulators of EMT. The study is well-designed and describes various molecular pathways involved, however there is some information missing and clarifications needed.

  1. It was reported that 75% showed a strong nuclear KDM5C staining. Please provide the percentages of positive tumor cells and staining intensity together with the type of distant metastasis data. I also recommend adding representative images of KDM5C staining in various types of distant metastasis. Were there any differences in the staining percentage and intensity between various distant metastases? Please provide this information.
  2. Did you also observe the correlation between E-cadherin and KDM5C (and inversed correlation between N-cadherin and KDM5C) areas in the clinical samples from patients? Could you provide this data together with IHC staining?
  3. As PC3 cells with KDM5C showed increased N-cadherin and decreased E-cadherin expression. Could you also provide N-cadherin staining on mouse xenografts of PC3 wild type and KDM5C knockdown cells in Figure 7.
  4. It is concluded that the reduced tumor growth in mouse xenografts is not a result of the differential proliferation of these cells as shown by the unaltered Ki-67 expression in PC3 and KDM5C knockdown cell However, the shDM5C1.6 knockdown showed p value of “*”, meaning that these cells have significantly increased Ki-67 index in Figure 7. Could you clarify this?
  5. Please provide staining of KDM5C protein in PC3 and shRNA-transduced cell mice xenografts as a control of depletion of this protein in the two conditions.
  6. It is supposed that tumor growth might be reduced due to changes in the tumor microenvironment after KDM5C knockdown. To support this hypothesis, did you observe any changes in the tumor micro-environment between the two conditions?

Author Response

Re:Manuscript ID: cancers-1663458

Title: Histone demethylase KDM5C drives prostate cancer progression by promoting EMT

Dear Mr. Xie,

dear editor,

dear rewiever,

thank you for having reviewed our manuscript entitled `Histone demethylase KDM5C drives prostate cancer progression by promoting EMT` by Lemster et al. and for being willing to re-consider a revised manuscript.

We would also like to thank the reviewers for their constructive criticism. We carried out additional experiments and were able to address criticism raised by the reviewers.

Please find a point-by-point response to the criticism raised by the editor and the two reviewers below.

We hope that our manuscript is now acceptable for publication in Cancers and look forward to hearing from you.

Sincerely,

Jutta Kirfel

Point-to-point letter addressing the reviewer 1 comments

Reviewer 1:

  • It was reported that 75% showed a strong nuclear KDM5C staining. Please provide the percentages of positive tumor cells and staining intensity together with the type of distant metastasis data. I also recommend adding representative images of KDM5C staining in various types of distant metastasis. Were there any differences in the staining percentage and intensity between various distant metastases? Please provide this information.

We thank the reviewer for this valuable comment. Nuclear KDM5C expression was analyzed by three experienced pathologists by eye balling and the complete cases were scored as negative, weak, moderate or strong (0, 1, 2, 3). As staining was very homogenous (see Figure 1 and S2), respective percentages of positive tumor cells were not investigated. Results were grouped in positive (1-3) vs. negative (0) cases.

  • Did you also observe the correlation between E-cadherin and KDM5C (and inversed correlation between N-cadherin and KDM5C) areas in the clinical samples from patients? Could you provide this data together with IHC staining?

Negative correlation between KDM5C and E-Cadherin expression in metastatic PCa patients is described in the text and shown in Supplementary Figure S2.

  • As PC3 cells with KDM5C showed increased N-cadherin and decreased E-cadherin expression. Could you also provide N-cadherin staining on mouse xenografts of PC3 wild type and KDM5C knockdown cells in Figure 7.

We agree with the reviewer, performed the N-cadherin staining and displayed N-cadherin staining on mouse xenografts of PC3 wild type and KDM5C knockdown cells in a new Figure 7. The expression of N-cadherin in knockdown xenografts is reduced up to 30% in comparison to those of PC3-cells. In addition, the expression of KDM5C, N-Cadherin and E-cadherin was analyzed by qRT-PCR (see below, point 5).

  • It is concluded that the reduced tumor growth in mouse xenografts is not a result of the differential proliferation of these cells as shown by the unaltered Ki-67 expression in PC3 and KDM5C knockdown cell However, the shDM5C1.6 knockdown showed p value of “*”, meaning that these cells have significantly increased Ki-67 index in Figure 7. Could you clarify this?

We thank the reviewer for this remark. We corrected the misleading Figure 7B, now showing that there is no significant difference in Ki-67 indices of PC3 wild type and knockdown xenografts (pooled results of both clones).

  • Please provide staining of KDM5C protein in PC3 and shRNA-transduced cell mice xenografts as a control of depletion of this protein in the two conditions

We thank the reviewer for this remark. Before injection of wild-type and stable KDM5C knockdown cells into the flank of the mice, expression of KDM5C was monitored.

Due to the time limitation for the revision (5 days), we now extracted RNA from the paraffin-embedded xenograft tumors using the Maxwell RSC RNA FFPE Kit together with the Maxwell RSC instrument (Promega, Fitchburg, WI, USA). Quantification was done with the Qubit fluorimeter (ThermoFisher, Waltham, MA, USA) and qRT-PCR reactions were performed for KDM5C, CDH1 and CDH2, respectively (Supplementary Figure S5).

  • It is supposed that tumor growth might be reduced due to changes in the tumor microenvironment after KDM5C knockdown. To support this hypothesis, did you observe any changes in the tumor micro-environment between the two conditions?

The reviewer raised an important question for further prospective experiments. The focus of this paper was to analyze the role of KDM5C in prostate cancer progression and metastasis. Our findings provide novel insights into the function of KDM5C that could be instructive for understanding of pathological processes.

Tumor microenvironment (TME) plays a significant role in understanding cancer progression and metastatic spread in PCa. The TME consists of different non-epithelial cell types including fibroblasts, immune cells, and endothelial cells (ECs), as well as ECM proteins such as collagen, laminin, fibronectin and hyaluronate. All these elements interact with tumor cells through a complex network of cell membrane receptors, cytokines, chemokines, growth factors and matrix remodeling enzymes.

Until now, we did not investigate the rate of total immune cell infiltration and/or determination of the expression of CD4 and CD8 to characterize the tumor microenvironment. It might also be better to use 3D culture systems for better recapitulating the in vivo organization and the tumor microenvironment or orthotopic xenografts for understanding the specific interactions of tumor cells and their organ microenvironment.

Reviewer 2 Report

The manuscript, “ Histone demethylase KDM5C drives prostate caner progression by promoting EMT” by Lemster et al, reports increased expression of KDM5C in metastatic prostate cancer. Utilizing human specimens and various cancer biology assays (cell culture and animal models), authors show that the KDM5C expression alterations promote prostate cancer progression through epithelial-mesenchyme transition mechanism.  The experiments are well designed and are of high quality.  The data are quite compelling.  

The authors need to address the following concern:

Although very well done, all the cell biologic data presented is based on one prostate cancer cell line, PC3. Unless there is serious limitation, it would be better to show some key data with additional 1-2 prostate cancer cell lines using siRNA and few EMT markers.

Author Response

Re:Manuscript ID: cancers-1663458

Title: Histone demethylase KDM5C drives prostate cancer progression by promoting EMT

Dear Mr. Xie,

dear editor,

dear rewiever,

thank you for having reviewed our manuscript entitled `Histone demethylase KDM5C drives prostate cancer progression by promoting EMT` by Lemster et al. and for being willing to re-consider a revised manuscript.

We would also like to thank the reviewers for their constructive criticism. We carried out additional experiments and were able to address criticism raised by the reviewers.

Please find a point-by-point response to the criticism raised by the editor and the two reviewers below.

We hope that our manuscript is now acceptable for publication in Cancers and look forward to hearing from you.

Sincerely,

Jutta Kirfel

Point-to-point letter addressing the reviewer 2 comment

Rewiever 2:

  • The authors need to address the following concern: Although very well done, all the cell biologic data presented is based on one prostate cancer cell line, PC3. Unless there is serious limitation, it would be better to show some key data with additional 1-2 prostate cancer cell lines using siRNA and few EMT markers.

PCa cell lines can be classified by different characteristics, including their site of origin, xenotransplant passage if applicable, histopathology, androgen dependency status and others. The focus of the paper was to analyze the role of KDM5C in metastasis. Only a few prostate cancer cell lines are commecially available like LnCAP (deived from lymph node metastases) and PC3 (from bone metastases). The cell lines ALVA-41 or ALVA-101 (from metastasis), which are similar to PC-3 cells, cannot be ordered by ATCC. As shown in Figure 2A KDM5C is strongly expressed in CRPC cells (PC3) and only moderately expressed in metastatic androgen-sensitive cells (LNCaP). Therefore, in our view it is not meaningful to repeat the experiments in LNCaP cells. Additionally, such further experiments are not achievable due to the time limitation of the revision process (5 days).